

# EXPLUME v1.0: a model for personal exposures to ambient O$_3$ and PM$_{2.5}$

Myrto Valari[1], Konstandinos Markakis[1], Emilie Powaga[2], Bernard Collignan[2], and Olivier Perrussel[3]

[1]LMD/IPSL, Laboratoire de Météorologie Dynamique, Sorbonne Université, Ecole Polytechnique, IPSL Research University, Ecole Normale Supérieure, CNRS, 75252 Paris, France
[2]Université Paris-Est, Centre scientifique et Technique du Bâtiment, Direction Santé Confort, Division Physico-chimie - Sources et Transferts de polluants
[3]AIRPARIF, Association de surveillance de qualité de l'air en Île-de-France, 7 rue Crillon, 75004, Paris, France

**Correspondence:** Myrto Valari (myrto.valari@lmd.polytechnique.fr)

**Abstract.** This paper presents the first version of the regional scale personal exposure model EXPLUME. The model uses simulated gridded data of outdoor O$_3$ and PM$_{2.5}$ concentrations and several population and building-related datasets to simulate 1) space-time activity event sequences, 2) the infiltration of atmospheric contaminants indoors and 3) daily aggregated personal exposures. The model is applied over the greater Paris region at 2km x 2km resolution for the entire 2017 year. Annual averaged
population exposures are discussed. We show that population mobility within the region, disregarding pollutant concentrations indoors, has only a small effect on average daily exposures. By contrast, considering the infiltration of PM$_{2.5}$ in buildings decreases annual average exposure by 11% (population average). Moreover, accounting for PM$_{2.5}$ exposure during transportation (in-vehicle, while waiting on subway platforms, and while crossing on-road tunnels) increases average population exposure by 5%. We show that the spatial distribution of PM$_{2.5}$ and O$_3$ exposures is similar to the concentration maps over the region,
but the exposure scale is very different when accounting for indoor exposure. We model large intra-population variability in PM$_{2.5}$ exposure as a function of the transportation mode, especially for the upper percentiles of the distribution. 20% of the population using bicycles or motorcycles is exposed to annual average PM$_{2.5}$ concentrations above the EU target value (25 $\mu$g/$m^3$), compared to 0% for people travelling by car. Finally, we develop a 2050-horizon projection of the building stock to study how changes in the buildings' characteristics to comply with the thermal regulations will affect personal exposures. We
show that exposure to ozone will decrease by as much as 14% as a result of this projection, whereas there is no significant impact on exposure to PM$_{2.5}$.

## 1 Introduction

Despite significant improvement in European air-quality over the past decades, in 2013 approximately 90% of the urban population was still exposed to concentrations exceeding the WHO Air Quality Guidelines for health (EEA, 2015). Several
epidemiological studies have shown the adverse health effects of exposure to PM$_{2.5}$ and O$_3$ (Cohen et al., 2005; Levy et al., 2005; Pope and Dockery, 2006). In the majority of these studies, the exposure surrogate associated with morbidity or mortality metrics is a spatially aggregated pollutant concentration from measurements at different sites over the urban agglomeration





(Anderson et al., 2004; Bell et al., 2005). The underlying hypothesis here is that exposure is homogenous over the population. For this assumption to be valid, these studies are limited to small geographical zones where population density and pollutant

concentrations are also homogenous (Sarnat et al., 2007). Several shortcomings of this approach have been raised in previous studies. On one hand, pollutant concentrations are spatially heterogeneous, especially within cities where different emission sources co-exist and the presence of buildings imposes barriers to the dispersion of pollutants. For example, a Health Effects Institute report states that the zones most impacted by traffic-related pollution are up to 300–500m from highways and other major roads, and when calculated for large cities in North America that affects 30 – 45% of the population (HEI, 2010). Intra-

urban variability in pollutant concentration is a principal source of exposure misclassification in environmental epidemiology models, leading to errors in the evaluation of the health risk (Blair et al., 2007; Edwards and Keil, 2017). Furthermore, large variability in population exposure arises from human activity, population mobility, transport usage, building characteristics (Georgopoulos et al., 2005). Therefore, to study the health risk on specific population groups — such as children, elderly people, asthma patients or pregnant women (Olsson et al., 2014) or the health effect of co-pollutants (Olstrup et al., 2019b, 

a; Valari et al., 2011), or the risk associated to living or working near busy roads (Lipfert and Wyzga, 2008; Miranda et al., 2013) — one has to account for pollutant concentration at district-level, population dynamics, and exposure indoors and during transport (Franklin et al., 2012; Hodas et al., 2012).

To answer this emerging demand, several methods for estimating personal exposure have been developed. Land-use regression models have been largely used to relate concentrations measured at monitor sites with concentration estimates at different

locations across the city (Beelen et al., 2013; Cattani et al., 2017; Ryan and LeMasters, 2007). Then, space-time activity data are coupled to concentration data to provide exposure estimates (Vizcaino and Lavalle, 2018; Xu et al., 2019a). Land-use regression models provide spatial maps where urban features such as roads, buildings, parks may be distinguished from background concentration levels. But concentration gradients resulting from the regression do not account for the dynamical or chemical processes taking place at these scales. Portable instruments, based on mass filters or high-accuracy optical methods

(reference sensors) have also been used during specific field campaigns to to measure exposure in cars, in subway trains and on subway platforms, and in residences or other indoor micro-environmental locations (Hwang and Lee, 2018; Lim et al., 2012; Morales Betancourt et al., 2019; Williams and Knibbs, 2016; Xu et al., 2019b). These methods are accurate but restricted to limited periods and spatial contexts. The availability of low-cost personal monitors (micro-sensors) is a new opportunity in the atmospheric exposure field. They provide access to almost real-time, high resolution concentration measurements (Xie

et al., 2017). However, the accuracy of these instruments, their calibration, as well as their high sensitivity to environmental conditions (e.g. humidity) and human manipulation are yet to be addressed before their true potential to be realized (Berchet et al., 2017).

Pollutant concentration fields simulated with atmospheric dispersion models are another possible input source for exposure models. The advantage of this approach is that simulation data may cover long time-periods to support climate studies or

policy applications adjusting for meteorological variability, emissions regulations, and land-use classification. Gaussian dispersion models such as ADMS-urban or AERMOD have often been coupled with population space-time activity data for use in exposure studies (Batterman et al., 2014; Kousa et al., 2002). These models, coupled with regional-scale chemistry-transport





models, account simultaneously for long-range transport, regional background concentrations, and local features such as traffic emissions over the road network (Soares et al., 2014).

Regional scale chemistry-transport models (CTMs) such as CHIMERE (Mailler et al., 2017) or CMAQ (Appel et al., 2014) have achieved resolution of 1km x 1km with sufficient accuracy to be considered for use in such fine scale applications (Beevers et al., 2013). Statistical, dynamical or hybrid downscaling techniques such as kriging (Beauchamp et al., 2015) or subgrid-scale parametrizations (Valari and Menut, 2010) can be applied or coupled to these models to provide concentrations at district level. The use of CTMs instead of high-resolution Gaussian or Lagrangian models in an exposure context has several advantages.

The study domain may be large enough to cover an entire region, whereas typical Gaussian or Lagrangian applications cover at best, the urban agglomeration. However, a large part of the population moves in and out of the agglomeration within the day and on systematic basis. Furthermore, the enhanced chemical mechanisms of CTMs compared to the simplified chemistry (the Chapman cycle) in Gaussian or Lagrangian models gives access to refined information on the chemical speciation and size distribution of particulate matter (PM). This information is particularly relevant in the context of health impact assessment,

since the health impact of PM strongly depends on these properties (Atkinson et al., 2015; Cassee et al., 2013).

This paper presents the first version of a regional scale model for personal exposures to $O_3$ and $PM_{2.5}$. Outdoor pollutant concentrations are simulated with the CHIMERE regional CTM at 2km x 2km resolution. The model selects sample populations that reproduce essential demographics at relevant geographical units (communes), namely age, gender, occupation, communes of residence and work, principal modes of transportation and construction dates of residence and workplace. Activity event

sequences for each member of the sample are developed by matching the distributions in the simulated population with distributions in the Enquête Globale de Transport (EGT, 2010) study. Infiltration of outdoor air-pollution indoors in dwellings, offices, and schools, is modeled with the SIREN model (Collignan et al., 2012), developed at the Centre Scientifique et Technique du Bâtiment (CSTB). For other indoor locations (cars, buses, subway and regional trains and tram) we apply indoor/outdoor ratios found in the literature from previous measurement campaigns conducted in the region. Adjustments are also applied for

specific activities such as cycling, walking on busy roads, waiting at the subway platforms, as well as for car journeys that intersect tunnels or the Boulevard Periphèrique (ring road). Space-time activity sequences define the geographical coordinates of each member of the population at each minute of the simulation. Daily averaged personal exposures are calculated from the products of time spent by a person in different microenvironments and the time-averaged pollutant concentrations occurring in those locations (Klepeis, 2006). Personal exposures are simulated for the entire 2017 year over the Ile-de-France region

(greater Paris).

## 2 Personal exposure calculation

We adopt the continuous formulation of exposure in space and time from Klepeis (2006):

$$E_i = \int\limits_{t1}^{t2} C_i(t,x,y,z)dt \tag{1}$$





where $C_i$(t,x,y,z) is the concentration at point (x,y,z) occupied by a receptor i at time t and $t_1$ and $t_2$ are the start and end times of a given exposure episode. As discussed there, the most accurate exposure assessment would rely on real-time personal monitoring devices affixed to people as they move within all the locations that are part of their daily routines. In practice, such equipment is too expensive to affix to large cohorts. Also questions such as the calibration of the monitors and the assessment of the uncertainties still need to be tackled before such studies could be carried out at regional scale.

In a modeling framework, discrete locations termed as microenvironments are considered rather than fully continuous space. Integrating in time the semi-continuous formulation applies (Klepeis, 2006):

$$E_i = \sum_{j=1}^{m} \int_{t_{j1}}^{t_{j2}} C_{ij}(t)dt \tag{2}$$

where $C_{ij}$ (t) is the concentration experienced by the receptor in the discrete microenvironment j at a particular point in time t over the time interval [$t_{j1}$, $t_{j2}$], where $t_{j1}$ is the start time for the exposure episode and $t_{j2}$ the end time. In equation 2 the exposure trajectory of the receptor is followed explicitly. This approach has been adapted in cohort studies such as McBride et al. (2007).

As in Klepeis (2006), in the exposure model developed here receptors are simulated through individuals. Further discretizing in time, we calculate exposure as the sum of the product of time spent by a person in different microenvironments and the time-averaged pollutant concentrations occurring in those locations:

$$E_i = \sum_{j=1}^{m} C_{ij}T_{ij} \tag{3}$$

Here $T_{ij}$ is the time spent in microenvironment j by person i with units in minutes, $C_{ij}$ is the air-pollutant concentration person i experiences in microenvironment j in units of [$\mu g/m^3$], $E_i$ is the integrated exposure for person i [$\mu g/m^3$ min], and m the number of different microenvironments. In this formulation, concentration $C_{ij}$ is averaged over the corresponding time period $T_{ij}$ .

The general structure of the model with the necessary input datasets for the exposure calculation is illustrated in Figure 1. Outdoor pollutant concentrations are simulated with a regional scale chemistry-transport model. We use hourly averaged data over a horizontal grid with 2km spacing in both the west-east and the south-north directions (Section 3.1). Indoors pollutant concentrations(in buildings and during transportation) are deduced from outdoor concentrations by applying indoor/outdoor ratios. The model does not account for indoor sources so far. For buildings, indoor/outdoor ratios are calculated through a ventilation model (Section 3.2.1). For other indoor microenvironments, indoor/outdoor ratios are either taken from previous studies in the Ile-de-France region or calculated from existing indoor and outdoor concentration data as is the case for subway platforms (Section 3.2.2).

To obtain activity event sequences that determine the location of each member of the simulated population in time we draw on the 2010 survey "Enquête globale de transport" (EGT, 2010) conducted by the Direction Régional et Interdépartemental de l'Equipement et de l'Aménagement d'Ile-de-France. This survey questioned 43 000 individuals and identified 143 000 journeys. Each journey is characterized by the origin and destination points, the motive for traveling, the duration and the means of





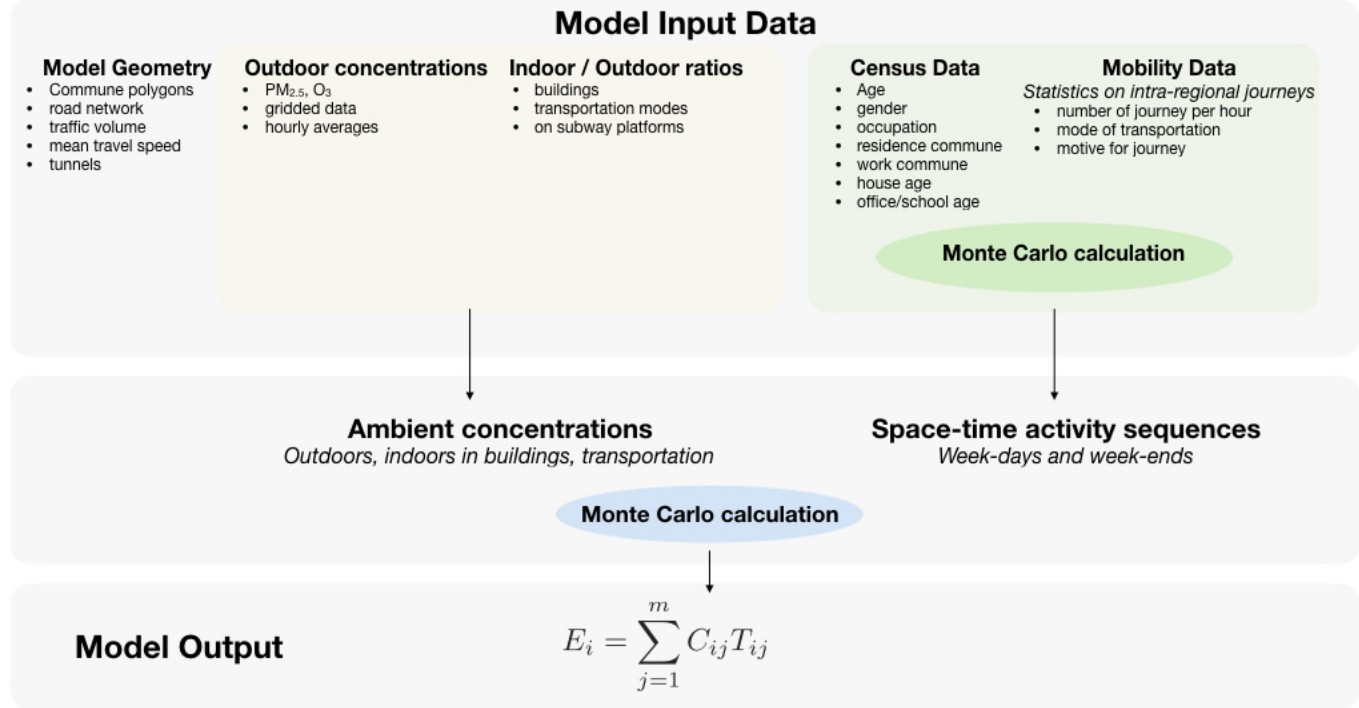

**Figure 1.** *Overview of the EXPLUME model structure, from the input data to the exposure calculation.*

transportation used. The mobility of the sample population is simulated with a Monte Carlo model that matches the simulated data with the EGT (2010) data (Section 4).

## 3 Pollutant concentrations

### 3.1 Outdoor $O_3$ and $PM_{2.5}$ concentration predictions

Pollutant concentrations are modeled with the CHIMERE model (Mailler et al., 2017) at a horizontal resolution of 2km x 2km. Four-level one-way nesting is used for the CHIMERE simulation with grids of 60, 20, 7 and 2km spacing between cells at both west-east and south-north directions. 15 vertical layers are used from 998 up to 300hPa with layers becoming thicker with distance from the surface level. Meteorological conditions are modeled with the Weather Research and Forecasting model (Skamarock et al., 2008) off-line at the same 4-level nesting grids as for the CHIMERE simulation but with a two-way nesting

configuration. Global HTAP (Hemispheric Transport of Atmospheric Pollutants) anthropogenic emissions are used outside the European continent, EMEP emissions for Europe outside the Ile-de-France region and finally a 1km x 1km resolution bottom up emission inventory developed by the AIRPARIF agency for anthropogenic emissions over the Ile-de-France region.



**Table 1.** Common measurements of statistical performance of the CHIMERE model, namely Pearson correlation, mean bias and root mean square error, aggregated over the whole 2017 year and for summer and winter seasons. Comparisons with traffic background and rural stations are conducted separately.

| | $O_3$ **hourly** | | $PM_{2.5}$ **hourly** | |
|---|---|---|---|---|
| | **Urban background** | **Rural** | **Urban background** | **Traffic** |
| **YEAR / Pears. Cor.** | 0.74 | 0.74 | 0.55 | 0.56 |
| **YEAR /Mean Bias [$\mu$g/m$^3$]** | 0.12 | 1.90 | 2.14 | -1.91 |
| **YEAR / RMSE [$\mu$g/m$^3$]** | 20.5 | 19.24 | 11.83 | 12.71 |
| **SUMMER / Pears. Cor.** | 0.72 | 0.75 | 0.19 | 0.23 |

Table 1 summarizes the comparison between the 2017-year simulation against measurements. A good temporal correlation on an hourly basis is observed for ozone, especially for the entire year (0.74) and summer periods (0.72). The correlation is lower for the winter period (0.61). The model overestimates ozone concentrations during summer, especially over rural stations, while a slight underestimation is observed in wintertime. This discrepancy is mostly due to night-time concentrations. CHIMERE predicts too low $O_3$ concentrations during night-time in winter, while model overestimates $O_3$ night-time concentrations during summer. This is probably due to overestimation in night-time NO concentrations in too shallow stable night-time residual layers at low winter temperatures, and consequently to enhanced $O_3$ titration by NO. For $PM_{2.5}$, model performance is significantly better for winter than for summer. Pearson correlation over urban sites estimated on an hourly basis drops from 0.58 (0.61 for traffic stations) for the winter period to 0.19 (0.23 for traffic stations) for summer. The CHIMERE model overestimates $PM_{2.5}$ concentrations over urban sites and underestimates them over traffic sites, showing the strong impact of traffic emissions on the $PM_{2.5}$ concentration over the urban area. The limitation of the 2km x 2km resolution in representing pollutant concentration near the road network is highlighted here.

Globally, we assume that the CHIMERE model at 2km x 2km resolution provides reliable $O_3$ and $PM_{2.5}$ background concentrations, being able to spatially differentiate the urban agglomeration from peri-urban and remote rural locations for $PM_{2.5}$. The formation of well-structured ozone plumes over the rural areas is also well represented (Figure 2). The model is also capable of reproducing the diurnal cycle of ozone and $PM_{2.5}$, especially near traffic sources where local emissions prevail. Pollutant episodes induced by favorable meteorological conditions are also well-captured by the model, even though a trend to underestimate ozone peaks and overestimate $PM_{2.5}$ peaks is observed.

Background $O_3$ and $PM_{2.5}$ concentrations for the exposure model are taken from the CHIMERE model simulation, with adjustments for concentrations over the traffic network and over the Boulevard Periphérique (road ring), where traffic emissions are particularly high and the model resolution fails to resolve these local effects.



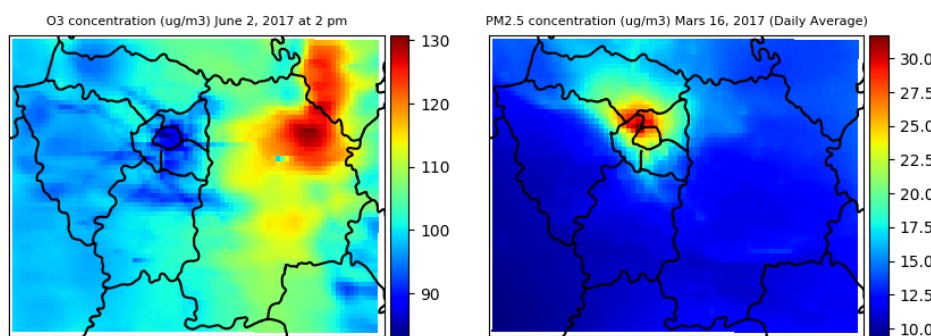

**Figure 2.** *Surface $O_3$ (left) and $PM_{2.5}$ (right) concentrations modeled with the CHIMERE chemistry-transport model.*

## 3.2 Infiltration of outdoor $O_3$ and $PM_{2.5}$ indoors

### 3.2.1 Dwellings, offices and schools

Indoor pollutant concentration levels depend on indoor sources and on outdoor pollutants entering the building through natural or mechanical ventilation. As air flows through the envelope of the building, pollutants react with the surfaces over which they flow. Therefore, the actual flow indoors depends on the specific path that the air flow takes: permeability of the building shell, natural air entry, or ducts (Walker and Sherman, 2013). Other sinks of pollutants indoor are deposition on the indoor

surfaces and chemical reactions with other indoor species. The relationship between these sources and sinks is expressed through equation 4 as in Walker et al. (2009).

$$\frac{dC_{X,in}}{dt} = \sum_i (P_{X,i} Q_{in,i}) \cdot C_{X,out} - (Q_{out} + \eta Q_h) C_{X,in} - k_d C_{X,in} - \sum_j k_j [chem_j] + \frac{C_X}{V} \tag{4}$$

Here,

- $C_{X,in}$ and $C_{X,out}$ are the concentrations of pollutant X indoors and outdoors respectively ($\mu g/m^3$)

- $P_{X,I}$ is the dimensionless penetration factor for the pollutant X through leak path i, i.e the fraction of the pollutant in the infiltration air that passes through the building shell or air entrance

- $Q_{in,i}, Q_{out}, Q_h$ are, respectively the volume-normalized air flow rates into the building through path i, out of the building, and through the heating ventilating and air-conditioning equipment expressed in Air Changes per Hour units ($h^{-1}$)





   – $\eta$ is the removal efficiency on the heating ventilating and air-conditioning equipment

– $k_d$ is the indoor deposition loss rate coefficient ($h^{-1}$)

   – $chem_j$ is the concentration of the $j^{th}$ chemical species reacting with the X pollutant ($\mu$g/m3)

   – $k_j$ the second order rate constant for the j reaction ($h^{-1}$)

   – $S_X$ is the time varying indoor production rate ($\mu$g/h)

   – V the volume ($m^3$)

Several studies have measured indoor/outdoor ratios for different building types and meteorological conditions, in cities around the world for ozone (Collignan et al., 2012; Weschler, 2000) and airborne particles (Cyrys et al., 2004; Matson, 2005; Monn, 2001). Results show a strong dependence on the building usage (residence or office/school), the air-tightness of the building, the ventilation system ,and the proximity to atmospheric pollution sources. Ozone I/O ratios generally vary between 0.2 and 0.7 (Weschler, 2000), while for $PM_{2.5}$, in the absence of indoor sources, between 0.5 and 1 (Morawska, 2003).

To account for the variability in I/O ratios due to these factors, we modelled ozone and fine airborne particles ($PM_{2.5}$) I/O ratios with the building ventilation model developed at the Centre Scientifique et Technique du bâtiment (CSTB), called SIREN (Collignan et al., 2012). The differential equation 4 is reformulated based on three assumptions: i) no indoor sources for $O_3$ and $PM_{2.5}$ ; ii) no chemical reactions with other atmospheric contaminants indoors; and iii) initial concentration indoors is null. We conducted simulations for a typical dwelling and office/school.

To account for the variability of I/O ratios due to air-tightness and ventilation systems, we applied a classification of the building stock based on the construction date. This information integrates air-tightness and ventilation systems evolution based on the national thermal and ventilation regulations (ADEME, 2013), the evolution of the building stock as described in (INSEE, 2014), and the use of ventilation systems in French buildings (OQAI, 2006). Table 2 shows the applied parametrizations for the different usages and construction dates.

We assume that the percentages of dwellings constructed before 1974 in the non-thermally rehabilitated and thermally rehabilitated classes are 25% and 75% respectively. No thermal rehabilitation is applied on offices and schools constructed before 1974. The air-tightness and ventilation systems for offices and schools built after 2012 do not change but the proportion of buildings in this category increases with time.

    Climatological conditions, temperature, pressure, and outdoor pollutant concentrations are simulated with atmospheric mod-
els (WRF for meteorology and CHIMERE for ozone and $PM_{2.5}$ concentrations) at a $4 \times 4 km^2$ horizontal resolution for a ten-year period from 1991 to 2000. Atmospheric fields are spatially averaged over the eight departments of the region. So, the atmospheric conditions database input for the SIREN model consists of ten-year period hourly data for the eight departments of the Ile-de-France region.

    For each Ile-de-France department eight SIREN simulations are conducted (five for dwellings and three for office/schools)
at a 3 min time-step. Penetration factor is fixed to 0.8 through the building shell and 1 through air inlet based on the state of





**Table 2.** Parametrization of the SIREN ventilation model depending on the construction date and the type of building, referring to the buildings' air-tightness and the ventilation system.

| | Dwelling | | Office/Scool | |
|---|---|---|---|---|
| | Air-tightness under 4 Pa depressurization | Ventilation | Air-tightness under 4 Pa depressurization | Ventilation |
| **Before 1974 not thermally rehabilitated** | 2.5 m³/h/m² | Natural ventilation based on the principle of rooms ventilated separately | 2.5 m³/h/m² | No ventilation system |
| **Before 1974 thermally rehabilitated** | 1.7 m³/h/m² | Natural ventilation based on the principle of rooms ventilated separately | NA | NA |
| **1974 - 2005** | 1.7 m³/h/m² | Cross ventilation principle induced by an exhaust mechanical ventilation system | 2.0 m³/h/m² | Cross ventilation principle by separated room induced by an exhaust mechanical ventilation system |
| **2006 - 2012** | 1.0 m³/h/m² | Cross ventilation principle induced by an exhaust mechanical ventilation system | 1.5m³/h/m² | Cross ventilation principle induced by a double flow mechanical ventilation system |
| **After 2012** | 0.6 m³/h/m² | Cross ventilation principle induced by a double flow mechanical ventilation system | NA | NA |

the art (Chen and Zhao, 2011; Monn, 2001; Stephens et al., 2012; Thatcher et al., 2003). Confronting numerical simulations with SIREN and I/O ratio measurements the deposition rate was fixed to 0.1 h$^{-1}$. The SIREN model output consists of a decade long database of I/O ratios for ozone and PM$_{2.5}$ at 3 min resolution for each of the eight Ile-de-France departments, for five construction date classes for dwellings and 3 construction date classes for offices and schools. This database is further

processed to provide seasonal I/O ratios for each pollutant, building type, construction date, and geographical zone as shown in Figure 3.



Open Access EGU

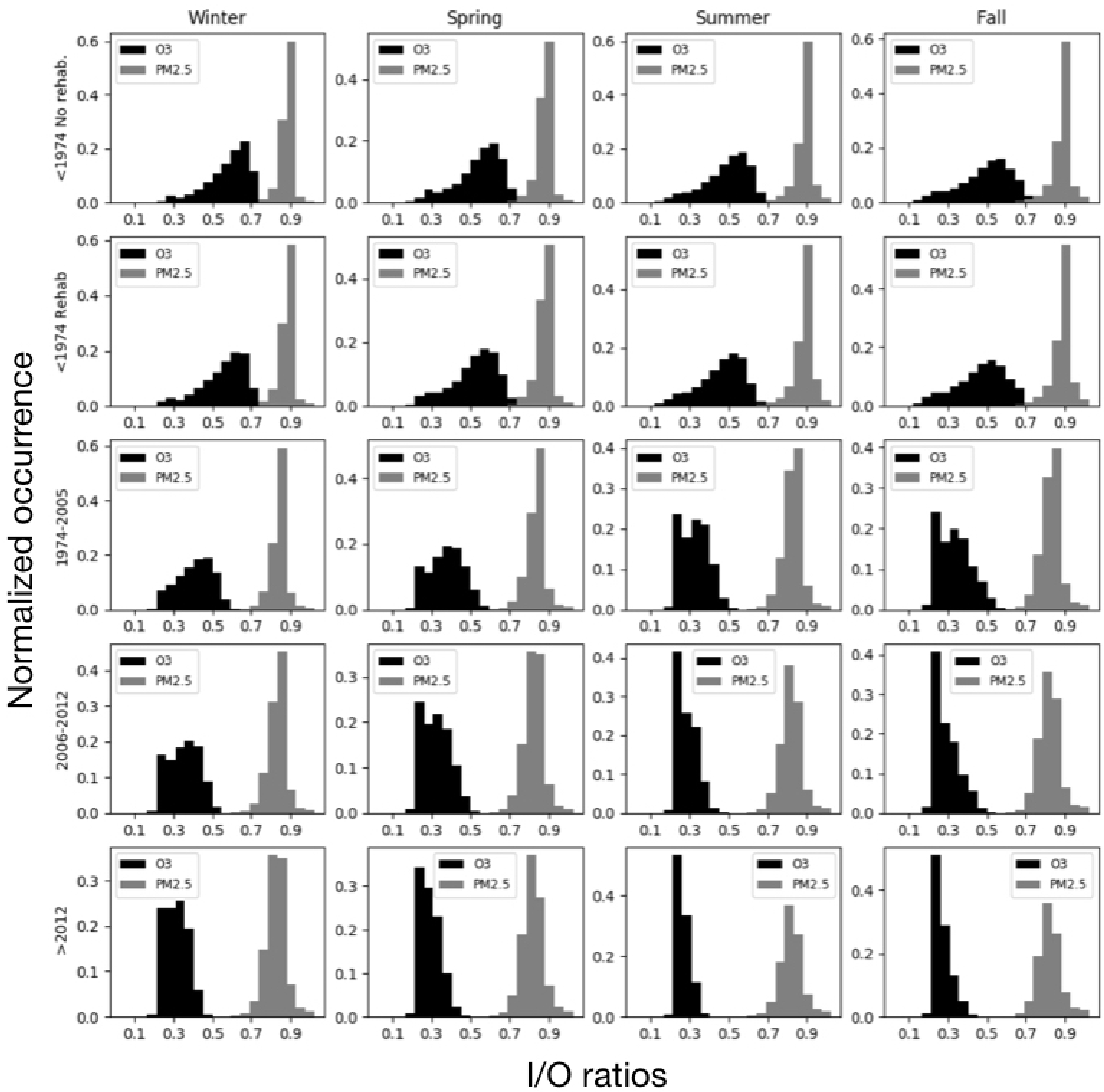

**Figure 3.** *$O_3$ and $PM_{2.5}$ indoor/outdoor ratios modeled with the SIREN model for dwellings (unitless). The number of occurrences of the histogram is normalized over the size of the dataset.*

### 3.2.2 Transportation

Ambient concentrations inside the principal transportation modes are deduced from outdoor concentrations by adjusting for indoor/outdoor coefficients taken from a study dedicated to evaluating the pollutant levels to which the Ile-de-France citizens are exposed while commuting to work and back during morning and evening rash hours (Delaunay et al., 2012). A significant number of contrasting situations is retained; twenty routes are chosen implementing the main modes of transport: car, bus,





subway, tramway, cycling and walking. Each route has been reproduced 30 times (15 round trips). The measurement campaign took place during the winter period of 2007 and 2008.

To define the indoor/outdoor ratio for each journey in the model we chose a random number within a uniform distribution
between the minimum and maximum values obtained by the study of Delaunay et al. (2012). The extreme values of these distributions are shown in Table 3. For public transport we distinguish between waiting on the platform and journey. For the suburban train (RER), we distinguish between journeys inside the subway network in the Paris agglomeration and the rest of the network. For journeys in cars, we distinguish between the road network in the Paris agglomeration, the Boulevard Periphèrique (road ring), and the rest of the network (rural).

**Table 3.** $O_3$ and $PM_{2.5}$ indoor / outdoor ratios for the principal means of transportation.

| | Waiting | | Journey | |
|---|---|---|---|---|
| | $O_3$ | $PM_{2.5}$ | $O_3$ | $PM_{2.5}$ |
| **Subway** | 1 | 1 | 0 | 1.7-3.7 |
| **Bus** | 0 | SQUALES | 0 | 5.5-8.5 |
| **Tram** | 0 | SQUALES | 0 | 5.5-8.5 |
| **On foot** | 0 | SQUALES | 0 | 5.5-8.5 |
| **Two wheels** | 0 | SQUALES | 0 | 5.5-8.5 |
| | **Paris intra-muros/outside** | | | |
| **RER** | 0/1 | SQUALES/1 | 0/0 | 3.2-5.4 / 2.9-3.2 |
| | $O_3$ | $PM_{2.5}$ | | |
| | | **Rural** | **Boulevard Periphérique** | **Paris agglomeration** |
| **Car** | 0 | 0.9-2 | 0.9-2.1 | 0.9-3.3 |

Several studies have shown that pollutant concentrations measured inside tunnels are several times higher than concentrations over the road but outside the tunnel. Orru (2015) conducted a study to evaluate the health impact of the exposure to traffic exhaust inside road tunnels. Here, we apply a special adjustment for car journeys that cross tunnels. We assume that if the itinerary of an individual intersects a grid cell (2km x 2km) containing a tunnel, there is a 20% probability that the driver will pass through the tunnel. In this case, we assume that the $PM_{2.5}$ concentration is two times higher than the outdoors
concentration (AIRPARIF, 2009).

$PM_{2.5}$ concentrations in the subway train tunnels are particularly high, especially for lines with rubber-tyred railway vehicles. To keep a record of the air-quality in the subway platforms the RATP (Régie Autonome des Transports Parisiens) operates measurements on a 24-hour basis at two metro stations and one RER platform (SQUALES). We used hourly on-platform





measurements of the SQUALES network and outdoor concentration measurements from the AIRPARIF network for the entire 2013 year to establish a diurnal cycle of the indoor /outdoor ratio inside the subway platforms (Figure 4).

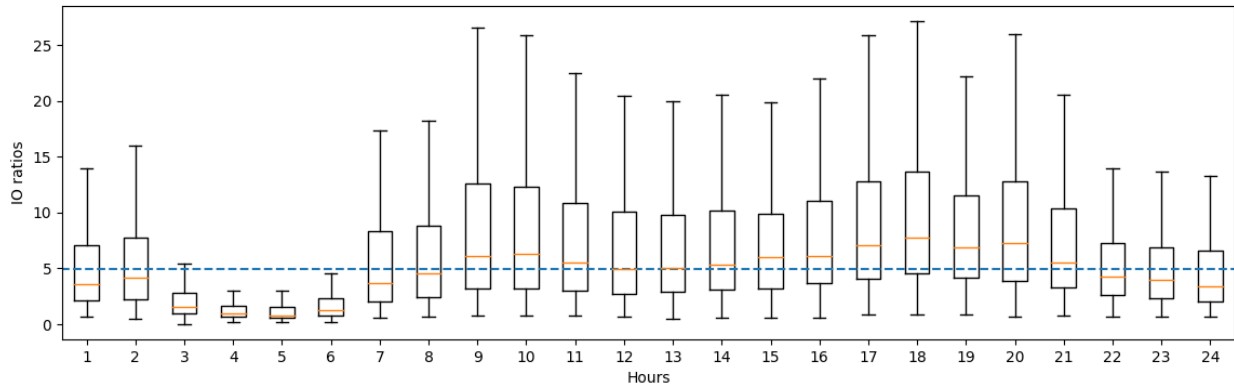

**Figure 4.** *Hourly distributions of indoor/outdoor PM$_{2.5}$ ratios for subway platforms issued from in-platform measurements of the SQUALES network and outdoor measurements of the AIRPARIF network. The blue line is the median I/O ratio.*


### 3.2.3 Other indoors

The SIREN model provides indoor /outdoor ratios for dwellings, offices and schools (Section 3.2.1). For other activities taking place indoors such as entertainment and shopping we use the same indoor/outdoor ratios that SIREN predicts for offices and schools. To decide whether shopping takes place indoors or outdoors we are based on statistics from the IAURIF (2006) study, following which 14% of the shopping activity in the Ile-de-France region takes place outdoors. Entertainment other than exercise is assumed to always take place indoors. For exercise activities, we first chose the type of exercise activity (IAURIF, 2006) and then whether it takes place indoors or outdoors depending on the specific activity.

### 4 Population data

The methodological steps to obtain activity event sequences for the sample population are listed here:

- select the population sample size that statistically reproduces essential demographics such as population of each administrative unit.

- assign attributes to the members of the population such as age, gender, principal occupation etc.

- simulate the mobility of the population by matching the journeys of the EGT (2010).





## 4.1 Generation of the sample population

The population data implemented in the model are public census data published by the INSEE (Institut Nationale de la Statistique et des Études Économiques). The current version of the model implements datasets for year 2009. The administrative unit chosen for the current study is the commune. The Ile-de-France region has 1300 communes and a population of 11 726 743. The most densely populated communes are located at the outer rings of the Paris agglomeration followed by a second circle of high population density at the suburbs directly attached to the agglomeration. A third highly urbanized ring is distinguished before reaching the rural areas at the outskirts of the Ile-de-France region (Figure 5).

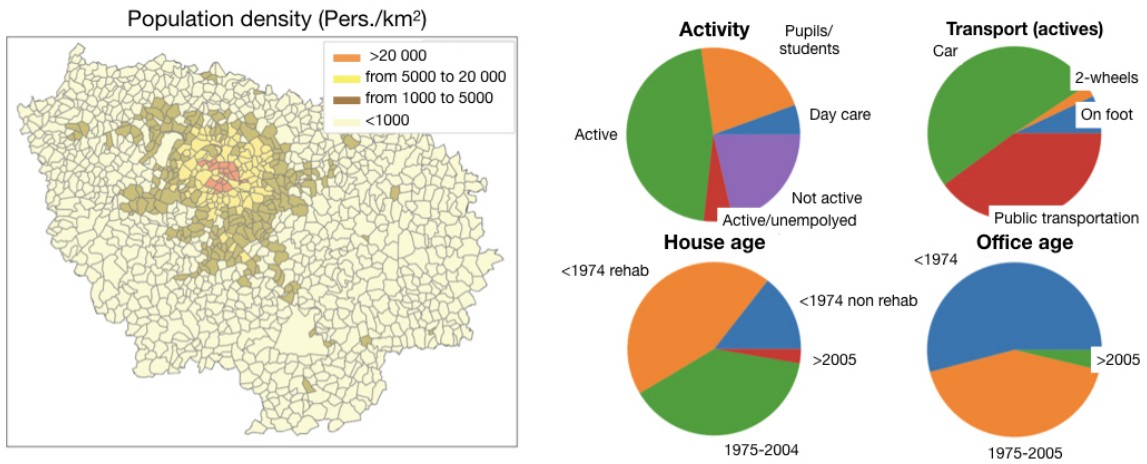

**Figure 5.** *Population density at the commune level in the Ile-de-France region (left). Distribution of exposure factors in the sample population (right). Data source census 2009 (INSEE).*


The size of the sample population is fixed at 250 000 individuals ($\approx$2% of the actual population). A sensitivity analysis showed that further increasing this number does not significantly affect the results of the simulation (not shown). The first module of the model sequence assigns several demographic attributes to each member of the sample population. These attributes, referred to as exposure factors hereafter, remain unchanged throughout the simulation. The procedure consists of
randomly selecting values from a distribution that matches the distribution of each attribute in the actual population of each commune. By repeating this process for each member of the sample population we are sure to reproduce the distribution of the exposure factors in the simulated population.

The population is divided in four age groups. Five occupation classes are defined, and two possible contract types (full-time or part-time) corresponding to 9-hours or 5-hours working days. Data on the construction date of the building of residence and
work are also implemented, to account for the infiltration of outdoor pollution indoors. The ten exposure factors implemented in the model are listed below. The different possible values for each model parameter are shown.

– **Home commune:** 1 out of 1300





- **Gender:** Male, female

- **Age group:** <4, 4-24, 25-64, >64

- **Occupation:** Day-care, pupils/students, active employed, active unemployed, not active (retired, at home, other)

- **Contract:** No contract, full-time, part-time

- **Work area:** Same commune as residence, different commune in the same department, different commune in different Ile-de-France department, outside IdF or abroad

- **Work commune:** 1 out of 1300

- **Means of transportation:** No transportation, on foot, 2-wheels, car, public transport

- **Construction date of residence:** <1974 not rehabilitated, <1974 rehabilitated, 1974-2004, >2005

- **Construction date of work place:** <1974, 1974-2004, >2005

A certain dependency exists between the exposure factors. For example, the professional occupation strongly depends on gender and age. To preserve the sub-population variability in the sample population, the random sampling of the exposure factors operates on stratified data, where exposure factors are supposed to be homogeneous. First, we assign the commune of residence, and then the other attributes in the following order: gender, age group, principal occupation, and finally the kind of contract. Once these primary attributes assigned, we then proceed to the selection of the rest, secondary characteristics. Working area is a function of the occupation and the commune of residence, the construction date of the buildings of residence depends on the commune, the gender and the age group. For offices general statistics are provided by ADEME for each Ile-de-France department dividing offices in three age classes.

## 4.2 Modeling the activity sequences

The second module of the model compiles 24-hour activity event sequences for each member of the sample population. Two diaries are compiled for each individual, one for weekdays and one for weekends. At each moment in time, people are either at home, engaged in an activity, or in transport. Eligible activities are the six motives for transport in the EGT (2010), namely work, professional affairs, school, market, recreation or personal affairs. From this study, we deduce the number of journeys to take place at each hour in the region for each of the six aforementioned motives. Whenever an activity ends, or once every hour if the person is at home, the model checks whether the individual is about to move. Some restrictions are implemented, because "not all individuals are eligible for all activities. For example, only certain age groups are eligible to go to day-care or school, only employed people are bound to go to work, etc. Once these restrictions are implemented, people will move in order to match the proportions of journeys per motive at each hour. If the person is bound to move, a number of choices are made in the following order (see also Figure 6):





- transportation mode

- destination commune

- travel distance

- travel time

- activity duration

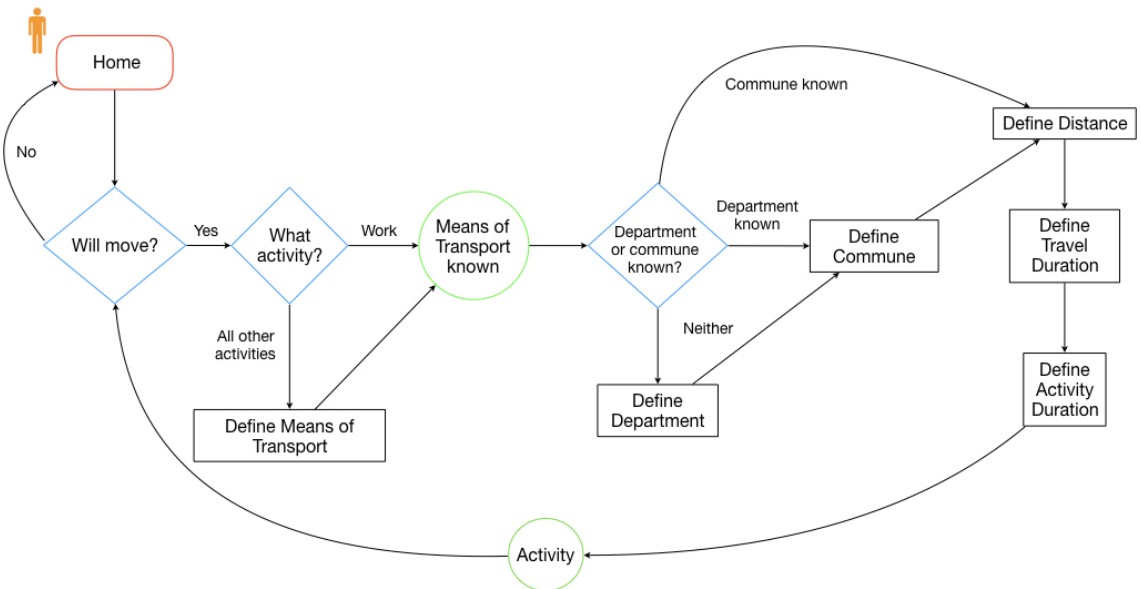

**Figure 6.** *Flow chart of the compilation of activity event sequences.*

For journeys to work and back, the INSEE provides a detailed dataset with the principal modes of transport. This information is part of the exposure factors assigned in the previous module (Section 4.1). The only stochastic choice here is for the 2-wheels case that has a 40% and 60% share between bicycles and motorcycles respectively (EGT, 2010). For the rest of the journeys 300 we match the proportions of the transportation modes per motive and hour from the EGT (2010).

In some cases, the destination commune is known (the person goes to work, to study, or back home). For other cases we only know whether the destination commune lies in the same department as the residence or in a different department. In this case, we first define the destination department based on data on the inter-departmental flows. Then we combine two pieces of information to assign the destination commune:

- data on the average distance travelled per means of transportation. We assume a straight line connecting the centroids of the communes. Several possible destination communes are selected based on the distance criterion.





– we use the information on the destination commune for the journeys in the EGT (2010) to assign a degree of attractiveness to the communes of the Ile-de-France region for each motive.

To assign the distance of the journey we distinguish between two cases. If the destination lies in a different commune, then the
journey distance is assumed to be equal to the distance over a straight line connecting the centroids of the two communes. If the destination lies within the commune of the current location then a stochastic choice is made for the traveled distance. We use statistics for the mean distance travelled per transportation means from the residents of the different departments. Depending on the transport mode, we assign a certain range around this average value and scale the limits to the commune size (radius of a circle with an area equal to the commune's area) and randomly chose a travel distance within this range.

To estimate the duration of the travel, we use two pieces of information at sub-communal scale. The first is the population density at $1x1km^2$ resolution. Individuals are distributed over the $1x1km^2$ resolution grid based on the population density. The second information is the average speed and flow over the road segments of the traffic network. We assume a straight line linking the centers of the origin and destination cells of the $1x1km^2$ grid and search for all grid-cells that intersect this trajectory. The speed at which the grid cell is passed through is assigned stochastically based on the distribution of speeds over
the road segments within the grid-cell. We note here that it would be more accurate to base our selection on the flows over each road segment rather than the speed distribution, but the geometry of the problem would become too complex. Given the high resolution of the application our insight is that this simplification is not bound to introduce significant errors to the transport model. The duration of the travel is then deduced from the distance and speed.

The final step is to define the duration of the activity. For children younger than 3 years old we use statistics on the time
spent at day-care. In all other cases we use statistics at department scale on the time spent by the population per activity.

A further distinction is whether the activity or transportation takes place indoors or outdoors. Certain activities may occur indoors or outdoors based on existing statistics (e.g. market and recreation). Possible means of transportation are on foot, two wheels (bicycle or motorcycle), car, bus, subway (metro), train (RER), and tramway. For public transportation, we distinguish between waiting time and travel time. For tramway, bus and RER outside Paris waiting takes place outdoors.

Figure 7 shows the results of the transport model. The diurnal patterns of the mobility of the population per motive are well reproduced. The model fails to reach the rush hour peaks, especially the morning peaks for work and school motives. It systematically underestimates the lunch hour peak. This is because the model does not implement secondary journeys i.e. people leaving the workplace to go for lunch and then back to work. These remarks become clear when looking at the total number of journeys (bottom of Figure 7), where we also see that the model underestimates the number of journeys at all hours.
However, the general picture of the simulation results is that the EGT (2010) data have been globally well implemented in the transport module.

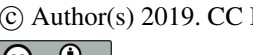



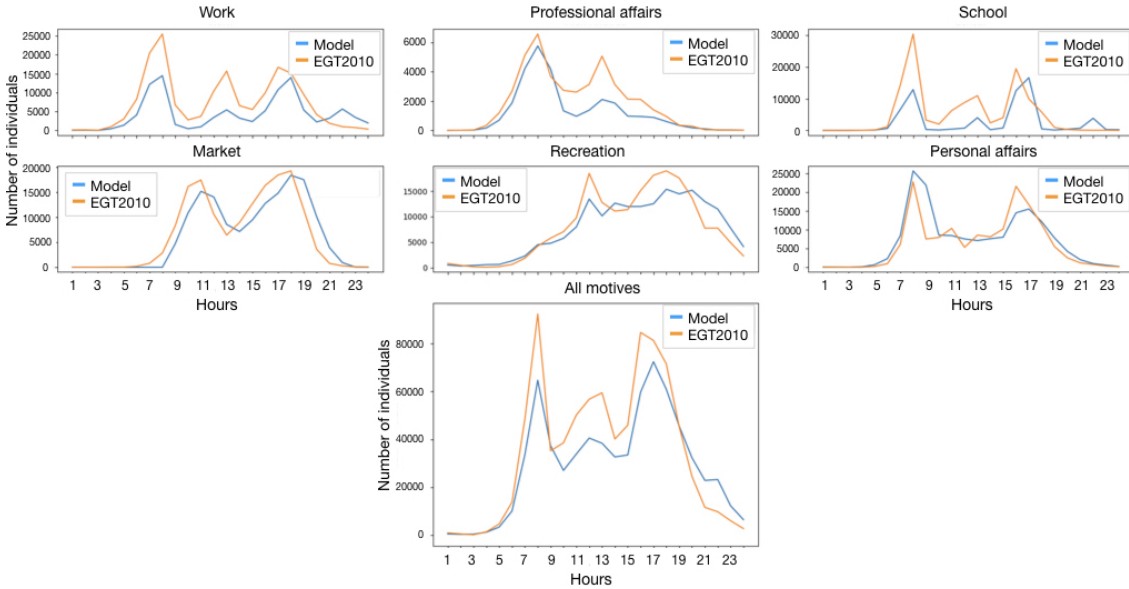

**Figure 7.** *Number of journeys per motive ($1^{st}$ and $2^{nd}$ rows) and total number of journeys all motives included (bottom) at each hour.*

The transport module simulates the mobility of the population. When individuals reach their destination they engage in the activity corresponding to the journey's motive. For activities other than travel we assign a mean duration. Activity event sequences are simulated at 1min temporal resolution. The ambient concentrations at which people are exposed during the

activity is the corresponding hour-averaged concentration modeled with the CHIMERE model at the grid-cell where the activity takes place. In the case of travelling, the model simulates the trajectory of the journey. For car journeys, we use the mean hourly traffic flows on each segment of the road network to assign probabilities to each road and assign the route trajectories. The trajectory of the journey may intersect several CHIMERE grid-cells. The corresponding outdoor concentrations are weighted by the time spent in each grid-cell to estimate the aggregated exposure. Figure 8 shows the number of people engaged in each

of the implemented activities at each hour of the simulation.





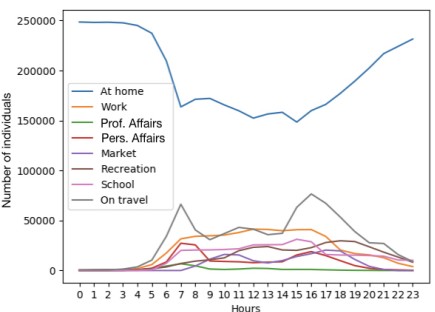

**Figure 8.** *Number of individuals engaged to the different activities at each hour of the simulation.*

## 5 Results

In this section we highlight different possible applications of the exposure model. The spatial distribution of exposure over the Ile-de-France region is discussed in Section 5.1, the relative contribution of each microenvironment in the daily aggregated exposure is quantified in Section 5.2, the variability in exposure patterns across sub-populations is studied in Section 5.3,

and the impact of considering 1) the infiltration of pollutants indoors and 2) the mobility of the population is illustrated in Section 5.4. Finally, in Section 5.5 we develop a 2050 horizon projection in the building stock of the Ile-de-France region, and quantify its impact in exposure to PM$_{2.5}$ and ozone.

### 5.1 Exposure maps

Personal exposures may be spatially averaged over the communes to provide exposure maps (Figure 9). The annual averaged

exposure to ozone is three times higher for the residents of the remote rural areas compared to the exposure of the Parisians. NO emitted by cars over the dense road network in the Paris city reacts fast with O$_3$ to form NO$_2$. This explains the absence of O$_3$ over the urban agglomeration. NO$_2$ emitted in large amounts over Paris under the influence of sunshine and in the presence of volatile organic compounds forms O$_3$ downwind, over the rural area (see also Figure 2). We also note that the exposure to ozone is much lower than outdoors ozone concentration (60 and 30 $\mu$g/m$^3$ background levels for the rural and urban areas

respectively (not shown)). This difference is due to the high amount of time people spent indoors, where ozone concentrations are close to zero (see also I/O ratios for O$_3$ Figure 3).

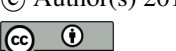

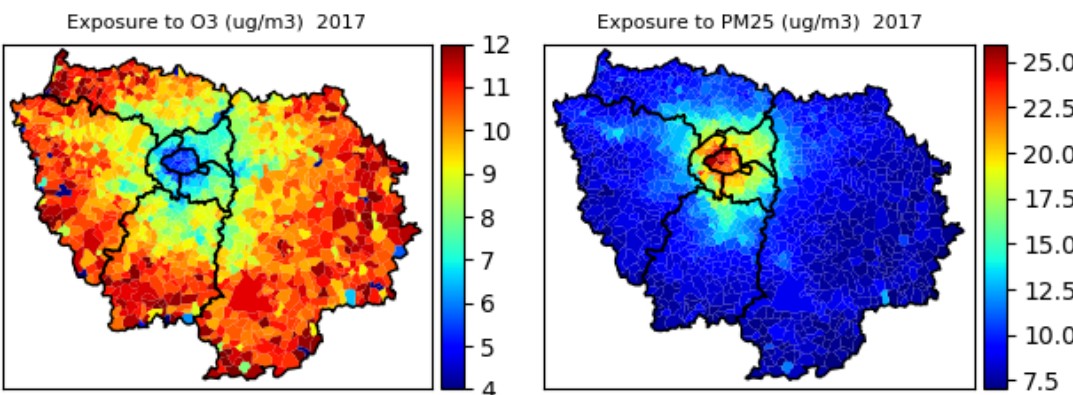

**Figure 9.** *Annual averaged $O_3$ and $PM_{2.5}$ exposure maps. Personal exposures are spatially averaged among the residents of each commune.*

The traffic network is a large source of $PM_{2.5}$, which explains why exposures to $PM_{2.5}$ are much higher in the Paris agglomeration than in the rural areas. Exposures to $PM_{2.5}$ are much closer to concentration levels because I/O ratios in buildings for $PM_{2.5}$ are closer to 1 than for $O_3$. Annual mean $PM_{2.5}$ concentrations are however lower than annual mean $PM_{2.5}$ exposures 365 (not shown). Even if indoor $PM_{2.5}$ sources in buildings are not yet implemented in the model and therefore concentrations in buildings are always lower than outdoor concentrations, $PM_{2.5}$ concentrations in cars, subway train or in subway platforms are several times higher than outdoor concentrations (see Table 3). Even if the time spent in transport is relatively lower than the time spent inside buildings, concentrations there are so high that the daily aggregated exposures are higher than outdoor concentrations. The construction date of buildings also plays an important role, with older buildings contributing to exposure 370 at much higher pollutant levels.




## 5.2 Exposure in different micro-environments

The relative contribution of exposure in different micro-environments in the aggregated daily exposure depends on outdoor concentrations, the indoor/outdoor coefficients if the activity takes place indoors, and the time spent in the micro-environment. Residential exposure accounts for almost 80% of daily exposure to both $O_3$ and $PM_{2.5}$ (Figure 10), reflecting the large amount
of time spent at home (see also Figure 8). Another 10% of daily exposure takes place at work. For ozone, exposure outdoors equals exposure at work. For $PM_{2.5}$ exposures in public transportation and car also play some role.

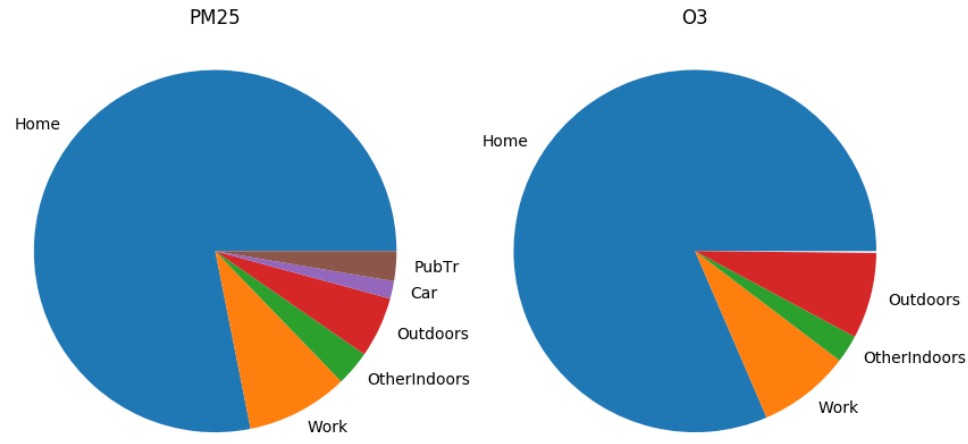

**Figure 10.** *Relative contribution of the different micro-environments in the aggregated daily exposures.*

## 5.3 Exposure of sub-population groups

Here we study the impact of several exposure factors on personal exposures. Figure 11 shows the cumulative distribution of exposure over specific sub-populations. We identify the two factors that have the largest impact on personal exposures, namely
the mode of transportation and the construction date of the building of residence. Both factors seem to strongly affect exposure to $PM_{2.5}$ and ozone. People traveling with motorcycles or cyces are exposed to the highest $PM_{2.5}$ levels, while exposure in cars is the lowest. 10% of the population using 2-wheels as transportation mode is exposed to $PM_{2.5}$ levels higher than the 25 $\mu g/m^3$ EU target value related to human health. The percentage of the population exposed to $PM_{2.5}$ levels above the EU target value drops to 5% for people travelling on foot, 3% for public transport and 1% for people travelling by car.
The construction date of the home building also plays an important role in personal exposure. For both pollutants, exposure is higher for buildings constructed before 1974. 100% of the population living in buildings constructed after 2005 are exposed





to PM$_{2.5}$ levels below the EU target value while 5% of the population living in constructions before 1974 is exposed to levels above the EU target value.

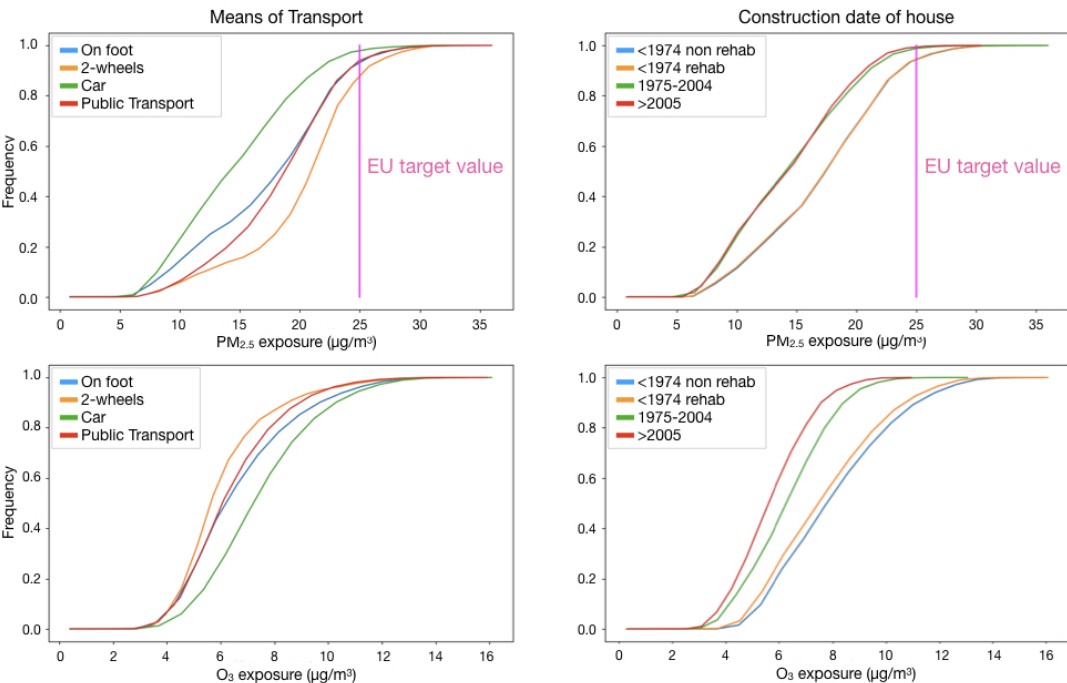

**Figure 11.** *Cumulative distributions of exposures to PM$_{2.5}$ (above) and O$_3$ (below) for sub-populations distinguished by the transportation mode (left) and construction date of the building where they live.*

### 5.4 Model sensitivity to population mobility and exposure indoors

We test the sensitivity of the calculation of personal exposures to PM$_{2.5}$ to the mobility of the population and the fact that exposure indoors is accounted for. For this, we conduct four simulations:

- REF The population stays at home and indoor concentrations are the same as outdoors.

- +MOBILITY The population moves but concentrations indoors are the same as outdoors.

- +INDOORS BUILDINGS The population stays at home and indoor / outdoor coefficients for buildings are applied.





– +INDOORS BUILDINGS & TRANSPORT The population moves and indoor / outdoor coefficients for both buildings
and transportation are applied.

Comparing the REF simulation with +MOBILITY shows that the mobility of the population within the region alone has
practically no impact on personal exposures (-1.5% on the median). This is partly explained by the relatively low spatial
resolution of the air-quality model simulation, but also by the fact that people spend most of their time indoors. Accounting

for residential exposure in the +INDOORS BUILDINGS simulation strongly affects personal exposures (-11% difference
with the REF in the median exposure). Accounting also for indoors exposure during transportation +INDOORS BUILDINGS
& TRANSPORT leads to a 4.6% increase in the median exposure compared to only accounting for residential exposures
(+INDOORS BUILDINGS). PM$_{2.5}$ concentrations during transportation are higher than outdoors whereas concentrations in
buildings are always lower than outdoors (no indoor sources in buildings). In the REF simulation 5% of the population is

exposed to concentrations above the EU target value of $25\mu g/m^3$ while in the complete implementation of indoor exposures
only 2% of the population is exposed to PM$_{2.5}$ above this threshold (Figure 12).

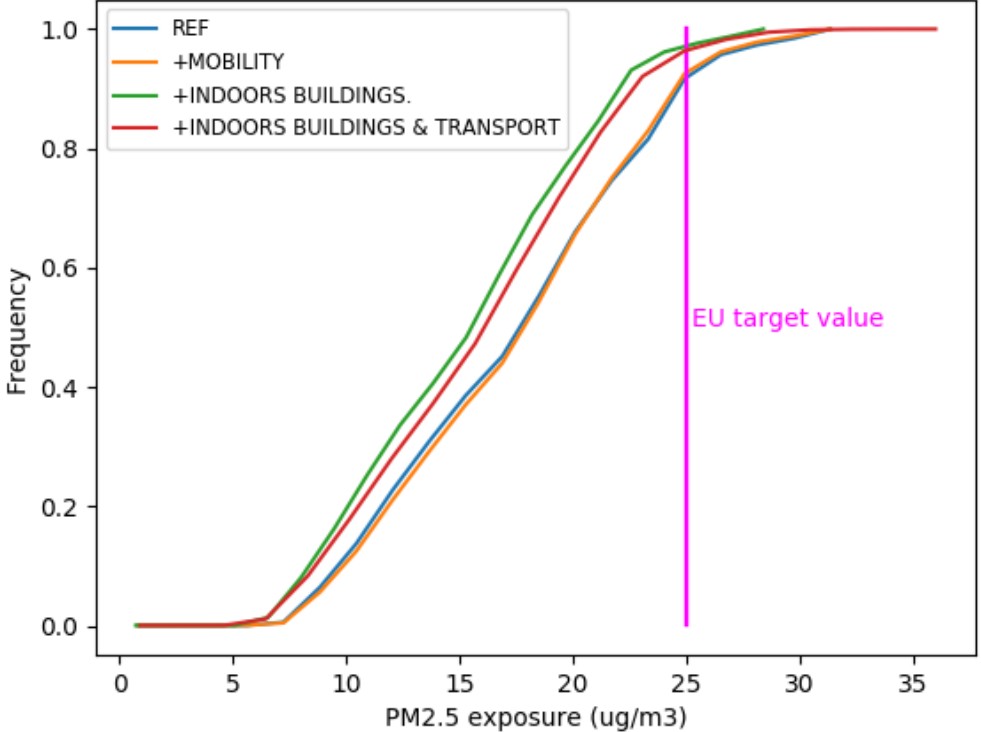

**Figure 12.** *Cumulative distributions of exposure to PM$_{2.5}$ resulting for simulations integrating increasing levels of complexity in the input data*





## 5.5  2050 horizon projection of the building stock

Based on data on the evolution of the French building stock (INSEE, 2014) and the national thermal building regulation found in the 2013 report of the Agence de l'Environnement et de la Maitrise de l'Energie (ADEME, 2013), the CSTB developed a
projection for the evolution of the building stock that is applied here for the 2050 horizon. To comply with thermal legislations and energy demand, buildings will tend to be more air-tight and ventilation systems more efficient. This evolution in the building stock will also affect air-quality in buildings, and therefore human exposure to atmospheric contaminants.

Following this projection, in 2050 dwellings, offices and schools will still fall in the same categories presented in Table 2 but the proportions of buildings falling in each category will change due to demolition, new construction and thermal rehabilitation.
The projection developed here models the annual rate of change in the building stock as follows:

For dwellings (Equation 5):

- Buildings belonging to the $5^{th}$ class (construction date >2012) will increase

- Buildings belonging in the $1^{st}$ class (<1974 not rehabilitated) will decrease due to demolition and thermal rehabilitation

- Buildings belonging to the $2^{nd}$ class (<1974 rehabilitated) will increase due to thermal rehabilitation of buildings in the
$1^{st}$ class

$$N_{D5}^{n+1} = N_{D5}^n + 0.012 * \sum_1^5 N_{Di}^n$$

$$N_{D1}^{n+1} = N_{D1}^n - 0.01 * \sum_1^5 N_{Di}^n$$

$$N_{D1}^{n+1} = N_{D1}^n - 0.015 * \sum_1^5 N_{Di}^n$$

$$N_{D2}^{n+1} = N_{D2}^n + 0.015 * \sum_1^5 N_{Di}^n$$

(5)

For offices and schools (Equation 6):

- Buildings belonging to the $3^{rd}$ class (2006-2012) will increase

- Buildings belonging to the $1^{st}$ class (<1974 not rehabilitated) will decrease due to demolition

$$N_{B1}^{n+1} = N_{B1}^n - 0.03 * \sum_1^3 N_{Bi}^n$$


(6)

$$N_{B3}^{n+1} = N_{B3}^n + 0.03 * \sum_1^3 N_{Bi}^n$$



The projection is applied to the Ile-de-France building stock, and we simulate personal exposures to quantify its impact. Due to new buildings being more air-tight with a better control of air renewal using more efficient ventilation systems, even less ozone penetrates the building shell. The resulting reduction in annual average exposure to $O_3$ is up to 14% (Figure 13). The change in annual averaged $PM_{2.5}$ exposure is very small (not shown).

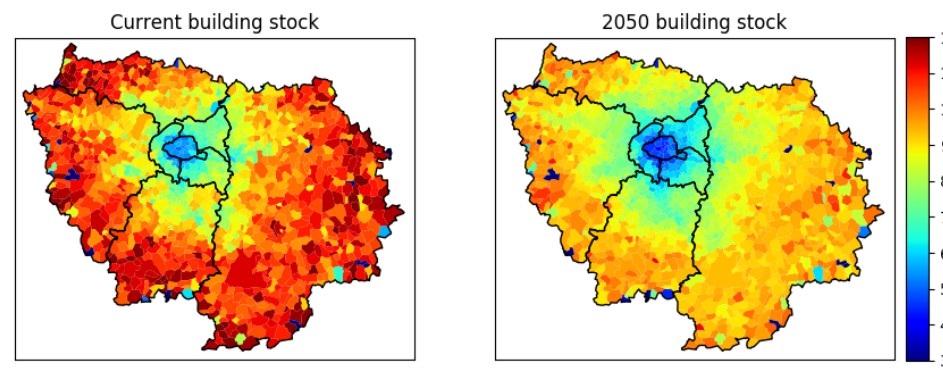

**Figure 13.** *Exposure to $O_3$ ($\mu g/m^3$) considering the actual building stock (left) and the 2050 horizon projection of the building stock (right).*

## 6  Conclusions


We developed a regional scale model for personal exposures to $PM_{2.5}$ and $O_3$. The model uses simulated outdoor pollutant concentrations and models the infiltration of outdoor contaminants indoors in buildings with a ventilation mass-balance model. Three building types are considered: dwellings, schools and offices. It also models population mobility inside the region considering the different possible transportation modes and adjusts for pollutant concentrations inside cars, buses, tram, subway

train and regional trains. A special treatment for concentrations in subway platforms is applied considering online measurements on the platform and outdoors. An adjustment for ambient concentrations inside road-tunnels is also applied from data from the literature. The model also uses data from the road traffic network to estimate the most probable trajectory for travel, as well as mean travel speed and duration.

We show that considering the population daily movement inside the region without accounting for the penetration of out-

door pollution indoors or indoor concentration during transportation has practically no impact on annual averaged personal exposures. This may be due to the coarse resolution of the outdoor concentration fields (CHIMERE simulation at 2km x 2km). However, if this resolution is not enough to solve concentration gradients at the proximity of local sources such as roads, it





is capable to distinguish between urban, suburban and rural concentrations. Most of the daily movement in the region crosses these boundaries (e.g. people living at the suburbs work in Paris and vice-versa).

On the contrary, accounting for the penetration of outdoor pollution indoors in buildings without considering population movement decreases annual averaged personal exposures by 11% for $PM_{2.5}$. This decrease stems only from the buildings' envelope acting as barrier to pollution infiltration indoors. When accounting also for population movement, annual averaged population exposures increase by 5%, showing the importance of exposure during transportation. Even if travelling represents only a small portion of time, exposures to $PM_{2.5}$ are too high and increase the daily burden of exposure.

We conclude that both infiltration of pollutant indoors and population movement need to be considered to estimate the aggregated daily exposure. We note here that, so far, the model does not implement indoor sources of $PM_{2.5}$ in buildings. We are aware that $PM_{2.5}$ indoors may be several times higher than outdoor concentrations (as is the case during transport). However, in this version of the model we were more interested to see how different building types and characteristics affect personal exposures independent of human activity that would drive indoor sources. The CSTB is working actively to develop

parametrizations accounting for indoor emission sources of $PM_{2.5}$ as well as their resuspension due to human activity.

    Several applications of the model are presented. We first show the maps of exposure to $O_3$ and $PM_{2.5}$ over the region. The spatial distribution of the exposure field is very similar to the concentration one, showing the strong correlation of the aggregated exposure to outdoor concentration. However, we show that if we focus on specific sub-population groups, such as people using bicycles or motorcycles systematically in their daily journeys or people living in houses built before 1974, the

upper percentiles of exposure are much higher than the general population. To study the impact of buildings' characteristics on personal exposures we implemented a 2050-horizon projection of the building stock in the Ile-de-France region. Following this projection, older buildings will be demolished or rehabilitated to comply with the thermal regulation and newer constructions will have modernized characteristics. The share of people living in the different building categories is modified to match this projection and personal exposures are simulated. 2050-horizon personal exposures to $O_3$ are decreased by as much as 14%

according to this projection.

    This first version of the model is parametrized for data available for greater Paris. However, the input data required for the simulation are also possible to find in other regions: census data, construction dates of buildings, mobility data. We can therefore imagine that with small adjustments in the format, the model could be applied to other regions. In all applications presented here, outdoor concentration data are simulated with the CHIMERE model and therefore the horizontal resolution is limited

to the order of 1km x 1km. However, this resolution limit is not inherent for the exposure model. If outdoor concentration fields at higher horizontal resolution were available from another dispersion model (e.g Gaussian or Lagrangian), the exposure calculation would have run without any modification being necessary.

*Code and data availability.*   The source code of the EXPLUME v1.0 model, as well as all necessary input data for the Ile-de-France region (open source see acknowledgements) are available under the doi:10.5281/zenodo.3352713



*Competing interests.* The authors declare that they have no conflict of interest.

*Acknowledgement.* This work has received funding from the European Union's Horizon 2020 research and innovation program under the grant agreement PULSE No. 727816 as well as from ANSES, ADEME, BelSPO, UBA and the Swedish EPA under the ERA-ENVHEALTH network grant agreement ACCEPTED no. 219337. This work was performed using HPC resources from GENCI TGCC under grant No. A0050110274. The authors acknowledge AIRPARIF for developing and providing the bottom-up anthropogenic emissions inventory used in

this study. We also acknowledge RATP for maintaining the SQUALES network and publishing the data as well as STIF, OMNIL and DRIEA for rendering public the EGT2010 dataset. Finally, we acknowledge Raphael Lachieze Rey for his valuable help in the statistical modeling.





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
