# Peer review of "EXPLUME v1.0: a model for personal exposures to ambient $O_3$ and $PM_{2.5}$"

_Geoscientific Model Development, 2019_

## Referee Comment (RC1) · Anonymous Referee #1 · 19 Nov 2019

General comments:

The manuscript describes a population-based human exposure model that simulates exposure to ozone and PM2.5 for a representative set of individuals and presents results from application of the model to the population of Paris, France for the year 2017. The model combines spatially and temporally varying ambient outdoor concentrations from an air quality model with data on population demographics, time-location sequences created for each simulated individual, and indoor/outdoor ratios for different types of buildings or transportation modes in a stochastic framework to estimate exposures. The modeling approach is consistent with previously developed population-based human exposure models for the U.S. and other European countries, but also takes advantage of available data for the region to advance certain aspects of the

approach including development of individual activity sequences that are defined geographically in space and time, and modeled seasonal distributions of indoor/outdoor ratios by building type and age. These features of the model support the originality of the work and make it of broader interest for publication. The manuscript should more directly note these unique aspects of the model and differences from previous approaches. In addition, output from the human exposure model were analyzed to examine the impact of certain factors on the exposure estimates including population mobility for daily activities (work, school, shopping), infiltration of air pollutants indoors, transportation mode, and projected future changes in building stock. These analyses demonstrate the utility of the model results to address a variety of research questions.

Specific comments:

1. Introduction - Line 20-25: epi studies cited are 10+ years old and for the U.S. only, cite current epi studies and at least some for Europe/France for relevancy. - Line 55-60: other references may be more appropriate e.g. review paper by Dias and Tchepel (2018) doi:10.3390/ijerph15030558 - Line 71 or 85: some motivation needs to be stated in this last paragraph of intro as to why the model was developed, what was the aim (e.g. for health impact assessment, epi study, etc.)? or just that now have essential components to combine in modeling approach? - Line 76-77: include important detail that SIREN model used to develop seasonal distributions of indoor/outdoor ratios for each type of building. This is a unique feature that could be highlighted as it is not reliant on measurements for a few locations that may not represent the area's buildings and people spend the majority of time in these microenvironments.

2. Personal exposure calculation - Line 90-93: repeated from intro so delete here or make 1 sentence for versus modeling. - Explanation of exposure calculations with 3 equations seems excessive; don't need first equation (general text is fine) and equation 3 is also in Fig. 1 so consider edits to reduce repetitive text. - Fig 1: could be improved . . . make more clear the two calculations of inputs (concentrations, population activity sequences) with different color boxes within first and second horizontal boxes (also

include 'microenvironment' with concentrations); add 'modeled' to 'PM2.5, O3' in first bullet under Outdoor concentrations, and 'buildings (modeled)' for first bullet under Indoor/Outdoor ratios; third box is really the population exposure calculation (as stated in figure title), include output as another box (maps, population distributions?); also would be helpful to include reference to Sections 3, 4 and 5 in figure for where described. - Although details on methods are generally well described, the Monte Carlo sampling aspects are not clear. Some attention is needed in methods to more clearly describe when and how Monte Carlo sampling is utilized vs. when a modeled value is used.

3. Pollutant concentrations - Modeled outdoor concentrations Model performance for the CHIMERE application (Table 1 and text) should include the number of measurement sites used and note the relative spatial coverage for each type of site. Would be helpful to include Box plots showing the distribution of the statistics across sites in supplemental material. Fig 2: Text and figure title should note that figures are maps of specific date/hour. Suggest including maps of annual average concentrations in same figure with these of the example hours. Line 148-149: Also include box plots comparing the diurnal trend in modeled and measured concentrations or in supplemental material. Since activity patterns have a diurnal pattern, it is important to understand how well the air quality model captures the diurnal pattern in concentrations. Line 151-153: is this adjustment what is described in 3.2.2 Transportation (Table 3)? Unclear why units for ozone are micrograms/m3, when parts per billion (ppb) is typical. - Indoor concentrations Fig 3: provide the actual distributions used as input to exposure calculations for all microenvironments in table similar to Table 3.

4. Population data - Line 243: add a concluding sentence that summarizes that Monte Carlo sampling method is used to randomly generate a data set of simulated individuals based on these steps - Line 288: not clear why quotes are used - Line 292-296: these could be a sentence list rather than bullet list. - Fig 8: figure is small and should be enlarged. Or add a few other small plots showing different examples (children <4 vs. adults 25-64)

5. Results - First sentence should be improved to convey that each of the sections looks at a different aspect of the model output as examples of its use in applications. - Line 354: add 'population' ... "to provide population exposure maps" - Line 369-370: sentence needs more explanation or refer to differences in indoor/outdoor ratios by age of buildings as done on line 361. - 5.2 could be improved by including comparisons such as between male/female and/or age groups in Fig 10.

6. Conclusions - Authors should compare/contrast the modeling approach or results with other similar work, e.g. Smith et al (2016) DOI: 10.1021/acs.est.6b01817, Shekar-rizfard et al (2016) http://dx.doi.org/10.1016/j.envres.2016.02.039

---

## Referee Comment (RC2) · Anonymous Referee #2 · 28 Nov 2019

I think the paper is interesting, but I have some concerns about it.

The main issue is that I found in the paper a lot of assumptions to be taken, to evaluate personal exposure. You would need in my opinion to careful explain the implication of these assumptions, and how these affect (or not) the robustness of your results. I.e.: - pag 6 row 150-155. The authors speak about 'adjustment' of the concentrations, without specifying what they are doing, and the implications in terms of results' sensitivity. - pag 8 row 190-195. Assumptions on dwelling are proposed (25% vs 75%) without explaining why and which are the implications of this choice - Table 2: how the modelling results change modifying these parameters? Are these parameters robust? - pag 11 row 220-225. "we assume that if the itinerary of an individual intersects...': also here I am not able to evaluate the impact of this assumption. The authors should in my

opinion clarify the impact of all these assumptions.

Another point. Now I already read lot of papers in which population activities are derived using GPS (mobile phone i.e.) data. This is a more direct approach to 'individual exposure' as it allows to get the mobility of the people (at least a proxy) with a reduced number of assumptions. You should at least explore this option in the paper, and explain pros and cons of your approach in comparison to a 'GPS data' based approach.

I would like (pag 6 row 144) a bit more on the CHIMERE validation. Now it is difficult for me to judge if the model is working properly on the selected domain. Also in view of the 'adjustment' that the authors have to do (see my previous comment).

In the introduction it seems that the authors are claiming that PM2.5 and O3 impacts on health are comparable. While in reality PM2.5 impact is much higher on population, than the O3 one. Please try to be more accurate here, quantifying these impacts (as from EEA report on Air Quality Status 2019, i.e.).

---

## Author Comment (AC1) · 7 Jan 2020

On behalf of all authors I would like to address a special thanks to the reviewer for helping us improve the paper. We appreciated and integrated all suggestions. You will find below a detailed report of all actions taken to address these comments. The line numbers refer to the annotated pdf where all changes with regards to the GMDD manuscript are tracked. This report is also attached here as supplement.

AC = Author comment

GENERAL COMMENTS

1) The modeling approach is consistent with previously developed population- based human exposure models for the U.S. and other European countries, but also takes

advantage of available data for the region to advance certain aspects of the approach including development of individual activity sequences that are defined geographically in space and time, and modeled seasonal distributions of indoor/outdoor ratios by building type and age. These features of the model support the originality of the work and make it of broader interest for publication. The manuscript should more directly note these unique aspects of the model and differences from previous approaches.

AC : We add a couple of sentences in the end of the introduction to highlight these original features and explain how the approach here is different from what is typically done in personal exposure assessment (lines 80-85).

SPECIFIC COMMENTS

Introduction

1.1 Line 20-25: epi studies cited are 10+ years old and for the U.S. only, cite current epi studies and at least some for Europe/France for relevancy.

AC : References on the health impact of atmospheric pollution have been updated in the introduction and eps studies for Europe an France in particular have been added (ines ∼ 20-25)

1.2 Line 55-60: other references may be more appropriate e.g. review paper by Dias and Tche- pel (2018) doi:10.3390/ijerph15030558

AC : We add the suggested reference to the review paper as well as references in there (line 64).

1.3 Line 71 or 85: some motivation needs to be stated in this last paragraph of intro as to why the model was developed, what was the aim (e.g. for health impact assessment, epi study, etc.)? or just that now have essential components to combine in modeling approach?

AC : We now explain the motivation behind the development of the personal exposure

model (line 83-85).

1.4 Line 76-77: include important detail that SIREN model used to develop seasonal distributions of indoor/outdoor ratios for each type of building. This is a unique feature that could be highlighted as it is not reliant on measurements for a few locations that may not represent the area's buildings and people spend the majority of time in these microenvironments.

AC : The information is now added in line 91.

2. Personal exposure calculation

2.1 Line 90-93: repeated from intro so delete here or make 1 sentence for versus modeling.

AC : We deleted these lines (lines 100-104).

2.2 Explanation of exposure calculations with 3 equations seems excessive; don't need first equation (general text is fine) and equation 3 is also in Fig. 1 so consider edits to reduce repetitive text. AC : We now include only the equation that is actually used for the exposure calculation. Some small edits are also done in the text to integrate the modifications and deliver a more direct message.

2.3 Fig 1: could be improved . . . make more clear the two calculations of inputs (concentrations, population activity sequences) with different color boxes within first and second horizontal boxes (also include 'microenvironment' with concentrations); add 'modeled' to 'PM2.5, O3' in first bullet under Outdoor concentrations, and 'buildings (modeled)' for first bullet under In- door/Outdoor ratios; third box is really the population exposure calculation (as stated in figure title), include output as another box (maps, population distributions?); also would be helpful to include reference to Sections 3, 4 and 5 in figure for where described.

AC : We would like to thank the reviewer for his/her suggestions to improve this Figure. We took them all into account (see Figure 1).
2.4 Although details on methods are generally well described, the Monte Carlo sampling aspects are not clear. Some attention is needed in methods to more clearly describe when and how Monte Carlo sampling is utilized vs. when a modeled value is used.

AC : We now add in sections 3 and 4 a sentence to explicitly state which variable is subject to random sampling.

Sub-section 3.1 lines 187-188 Sub-section 3.2.1 lines 245-250 Sub-section 3.2.2 lines 272-273 Sub-section 3.2.3 line 278

3. Pollutant concentrations

3.1 Modeled outdoor concentrations Model performance for the CHIMERE application (Table 1 and text) should include the number of measurement sites used and note the relative spatial coverage for each type of site.

AC : The number of measurement sites is included in Table 1. A discussion on the relative spatial coverage of each type of site is now discussed in lines 145-150.

3.2 Would be helpful to include Box plots showing the distribution of the statistics across sites in supplemental material.

AC : This done in Figure S2 (supplemental material). The figure is discussed in lines 173-175.

3.3 Fig 2: Text and figure title should note that figures are maps of specific date/hour. Suggest including maps of annual average concentrations in same figure with these of the example hours.

AC : This Figure (now Figure 3) has been modified to address these issues. New panels with annual averaged surface concentrations of O3 and PM2.5 have been added. The corresponding text is also changed to note that left panels show surface concentration maps at specific date/hour (lines 176-177).

3.4 Line 148-149: Also include box plots comparing the diurnal trend in modeled and measured concentrations or in supplemental material. Since activity patterns have a diurnal pattern, it is important to understand how well the air quality model captures the diurnal pattern in concentrations.

AC : This new Figure has been added in section 3.1 (Figure 2) and discussed in lines 155-165.

3.5 Line 151-153: is this adjustment what is described in 3.2.2 Transportation (Table 3)?

AC : Model evaluation shows clearly that PM2.5 concentration over traffic measurement sites are underestimated by the CHIMERE model by a factor of ∼2. At this stage of development of EXPLUME, to assess this bias, we assume that when an individual travels along the road network or is in a car that passes through a road-tunnel we multiply the PM2.5 concentration level by 2. We are currently working on the development of a subgrid-scale model that will account implicitly for these fine-scale effects. A more complete explanation of this issue is now given in lines 180-190.

3.6 Unclear why units for ozone are micrograms/m3, when parts per billion (ppb) is typical.

AC : The reason why ozone concentration units were in $\mu$g/m3 is that AIRPARIF measurements are in this unit. We now changed ozone concentration and exposure units to ppb throughout the text. This applies to all figures and model scores (bias and RMSE) in Table 1.

3.7 Indoor concentrations Fig 3: provide the actual distributions used as input to exposure calculations for all microenvironments in table similar to Table 3.

AC : As noted in the paper (lines 257-262) these distribution are uniform. All information is therefore provided through the minimum and maximum values given in Table 3. We do not see how a Figure of these distribution would add value in the presentation of the

model.

4. Population data

4.1 Line 243: add a concluding sentence that summarizes that Monte Carlo sampling method is used to randomly generate a data set of simulated individuals based on these steps

AC : This sentence is now added in Section 4 line 289.

4.2 Line 288: not clear why quotes are used

AC : This was just a typo.

4.3 Line 292-296: these could be a sentence list rather than bullet list.

AC : This has been now changed (line 340).

4.4 Fig 8: figure is small and should be enlarged. Or add a few other small plots showing different examples (children <4 vs. adults 25-64)

AC : What we found more relevant here, is to add this figure as a panel to the previous figure (now Figure 8).

5. Results

5.1 First sentence should be improved to convey that each of the sections looks at a different aspect of the model output as examples of its use in applications.

AC : We add this sentence at line 401-402.

5.2 Line 354: add 'population' . . . "to provide population exposure maps"

AC : OK (line 409)

5.3 Line 369-370: sentence needs more explanation or refer to differences in in-door/outdoor ratios by age of buildings as done on line 361.
[Figure]

AC : We explained this aspect more in detail in lines 424-427

5.4 5.2 could be improved by including comparisons such as between male/female and/or age groups in Fig 10.

AC : We added pie charts for different age groups to improve this section (Figure 10). The text has been modified to integrate the added information (lines 429-437).

6. Conclusions

Authors should compare/contrast the modeling approach or results with other similar work, e.g. Smith et al (2016) DOI: 10.1021/acs.est.6b01817, Shekarrizfard et al (2016) http://dx.doi.org/10.1016/j.envres.2016.02.039

AC : We would like to thank the reviewer for providing these references. We now discuss our results in comparison to the findings presented there. This improved significantly not only the presentation of our conclusions but also the sensitivity analysis of section 5.4 that was directly comparable to these previous studies. We integrate this comparison in section 5.4 (lines 450-475) and in the Conclusions (Section 6 lines 510-530).

Please also note the supplement to this comment:
https://www.geosci-model-dev-discuss.net/gmd-2019-259/gmd-2019-259-AC1-supplement.pdf

---

## Author Comment (AC2) · 7 Jan 2020

On behalf of all authors I would like to thank the reviewer for his/her interesting comments and suggestions. We addressed all the concerns raised. Please find below our detailed report with all actions undertaken to improve the manuscript accordingly. The number of lines correspond to the annotated manuscript with all changes with regards to the GMDD submission tracked. The report is also attached as separate file (supplement pdf).

AC = Author comment

GENERAL COMMENT

1. You would need in my opinion to careful explain the implication of these assump-

tions, and how these affect (or not) the robustness of your results.

AC : We address this issue carefully throughout the text (see our response to the reviewer's specific comments)

SPECIFIC COMMENTS

1. pag 6 row 150-155. The authors speak about 'adjustment' of the concentrations, without specifying what they are doing, and the implications in terms of results' sensitivity.

AC : This issue has been also raised by the 1st reviewer (see our detailed response in the Specific Comment # 3.5.

2. pag 8 row 190-195. Assumptions on dwelling are proposed (25% vs 75%) without explaining why and which are the implications of this choice

AC : This choice is based on statistical data derived fro the INSEE (2014) tables. This is said now more clearly in lines 228-230.

3. Table 2: how the modelling results change modifying these parameters? Are these parameters robust?

AC : The choice of the parameters presented in Table 2 is based on data from various official sources already given in the paper (lines 221-225) such as INSEE (2014) and ADEME (2013) and OQAI (2006). The sensitivity of the model output to these parameters had been indeed studied through sensitivity analysis. This preliminary work was not mentioned in the paper. We add some information in lines 225 - 230.

4. pag 11 row 220-225. "we assume that if the itinerary of an individual intersects...': also here I am not able to evaluate the impact of this assumption. The authors should in my opinion clarify the impact of all these assumptions.

AC : Several measurement campaigns have shown that PM2.5 concentrations inside road tunnels are several times higher than on the road outside the tunnel. Here we

used data from the AIRPARIF (2009) study to derive a coefficient to account for the increased exposure inside the tunnels. The model resolution is too coarse to resolve the road network and therefore explicitly account for the fact that a given trajectory intersects a road-tunnel. For this reason we assign a probability for this to happen. At this stage of development of the EXPLUME model we assign an arbitrary probability of 0.2. This number seems reasonable given the area of the model grid-cells and a typical length of a road tunnel in the Paris area. To be more accurate we need to analyse road traffic data. This is an interesting suggestion for future improvement of the EXPLUME model. We now explain this in lines 267-270.

5. Another point. Now I already read lot of papers in which population activities are derived using GPS (mobile phone i.e.) data. This is a more direct approach to 'individual exposure' as it allows to get the mobility of the people (at least a proxy) with a reduced number of assumptions. You should at least explore this option in the paper, and explain pros and cons of your approach in comparison to a 'GPS data' based approach.

AC : We would like to thank the reviewer for pointing to this omission. We now add a discussion on the challenges of using data from smart phones with built-in GPS in personal exposure assessment. We also explain how a combination of questionnaires-based data (activities and transportation modes) with trajectories derived from analysed GPS data would help assessing large part of the uncertainties related to the time-activity data in exposure science (lines 395-400).

6. I would like (pag 6 row 144) a bit more on the CHIMERE validation. Now it is difficult for me to judge if the model is working properly on the selected domain. Also in view of the 'adjustment' that the authors have to do (see my previous comment).

AC : The section of CHIMERE model validation (Section 3.1) has been significantly changed to assess the specific comments 3.1 to 3.6 of the 1st reviewer. We are confident that the new version provides enough information to judge the CHIMERE model performance over the study domain and the simulation period.

7. In the introduction it seems that the authors are claiming that PM2.5 and O3 impacts on health are comparable. While in reality PM2.5 impact is much higher on population, than the O3 one. Please try to be more accurate here, quantifying these impacts (as from EEA report on Air Quality Status 2019, i.e.).

AC : The epi studies in the Introduction have been updated in this revised version of the manuscript. The effects of ozone and PM2.5 are now addressed separately to highlight that the health impact of PM2.5 is higher (lines 20-27).

Please also note the supplement to this comment:
https://www.geosci-model-dev-discuss.net/gmd-2019-259/gmd-2019-259-AC2-supplement.pdf

───────────────────────────